# Grain structure control during metal 3D printing by high-intensity ultrasound

C.J. Todaro [1], M.A. Easton [1], D. Qiu [1], D. Zhang[1], M.J. Bermingham [2], E.W. Lui [1], M. Brandt [1], D.H. StJohn [2] & M. Qian[1]*

Additive manufacturing (AM) of metals, also known as metal 3D printing, typically leads to the formation of columnar grain structures along the build direction in most as-built metals and alloys. These long columnar grains can cause property anisotropy, which is usually detrimental to component qualification or targeted applications. Here, without changing alloy chemistry, we demonstrate an AM solidification-control solution to printing metallic alloys with an equiaxed grain structure and improved mechanical properties. Using the titanium alloy Ti-6Al-4V as a model alloy, we employ high-intensity ultrasound to achieve full transition from columnar grains to fine (~100 μm) equiaxed grains in AM Ti-6Al-4V samples by laser powder deposition. This results in a 12% improvement in both the yield stress and tensile strength compared with the conventional AM columnar Ti-6Al-4V. We further demonstrate the generality of our technique by achieving similar grain structure control results in the nickel-based superalloy Inconel 625, and expect that this method may be applicable to other metallic materials that exhibit columnar grain structures during AM.

[1] Centre for Additive Manufacturing, School of Engineering, RMIT University, Melbourne, VIC 3000, Australia. [2] Centre for Advanced Materials Processing and Manufacturing (AMPAM), School of Mechanical and Mining Engineering, The University of Queensland, St Lucia, QLD 4072, Australia. *email: ma.qian@rmit.edu.au

Fusion-based metal additive manufacturing (AM) processes are featured by small melt pools and steep temperature gradients from the solid–liquid interface toward the liquid metal. As a result, the solidification process shows a strong epitaxial growth tendency from layer to layer while the number of nucleation events is limited due to both the absence of potent nucleant particles and the small melt pool volume (consumed quickly by epitaxial growth). This leads to columnar grains along the build direction in most additively manufactured metallic materials, which cause property anisotropy, reduce mechanical performance and increase tendency toward hot tearing. Therefore, a key objective for metal AM is to replace coarse columnar grains with fine, equiaxed grains throughout the part[1–4].

The titanium alloy Ti-6Al-4V is the benchmark alloy of the titanium industry and the most extensively studied alloy for metal AM[5]. In fact, it has essentially been used as a yardstick for assessing the capability of each metal AM process developed to date[6]. However, Ti-6Al-4V fabricated by different fusion-based AM processes exhibits a strong columnar grain structure[7–10]. The columnar prior-β grains in AM-fabricated Ti-6Al-4V feature strong <001> orientation along the build direction. This gives rise to a β → α transformation texture[8,11–15], which is an important concern for AM qualification[16,17] because of the resulting anisotropy of mechanical properties[14,15,18–20]. In addition, the coarse columnar prior-β grains may further degrade the strength of Ti-6Al-4V according to the Hall–Petch relationship established for lamellar α–β Ti-6Al-4V[21–23] (exceptions can exist[24]).

Introducing potent nucleant particles by inoculation can realize the columnar-to-equiaxed transition by the Hunt criterion[25]. Varying the AM process parameters to change the thermal gradient $G$ and growth velocity $V$ in the melt pool has the potential to achieve equiaxed grains as well[7]. However, the low $G$ values required for equiaxed solidification of metallic alloys are not easily encountered during AM. For example, according to the $G$–$V$ plots established for Ti-6Al-4V[7] and Inconel 718 (ref. [26]), it requires one or two orders of magnitude lower $G$ values to realize equiaxed solidification of each alloy during AM. The combination of nucleant particles with process control can enlarge the equiaxed region on the $G$–$V$ plot. This has proved particularly effective for AM of Al-based metals via the addition of $Al_3$(Sc, Zr)[27], $TiB_2$ (ref. [28]), $Al_3Zr$ (ref. [1]) and TiC[29] nucleants. Unfortunately, it remains challenging to find a stable and potent nucleant for many commercially important alloys. Ti-6Al-4V is one such alloy. In fact, the introduction of foreign nucleant particles unavoidably changes the chemistry and cleanliness of the alloy. In addition, if the nucleant particles agglomerate together in the liquid metal to form clusters, which is difficult to completely avoid, significant undesired side-effects or consequences can occur in subsequent processing or demanding applications. In that regard, achieving fine equiaxed grains without the assistance of nucleant particles is preferred if practical.

The application of high-intensity ultrasound to crystallization from liquid to solid can noticeably affect the properties of the crystalline material[30]. Ultrasonic irradiation of liquids can cause acoustic cavitation: the formation, growth and implosive collapse of bubbles, which occurs instantly in molten metals (0.00003 s) by recent ultrafast in situ synchrotron X-ray imaging of the process[31]. Bubble collapse emits intense, localized shock waves of temperatures of ~5000 °C, pressures of ~100 MPa (1000 bar) and heating and cooling rates of >$10^{10}$ °C s$^{-1}$ (ref. [32]). Acoustic cavitation during solidification of metal systems agitates the melt to activate nuclei naturally present in the alloy[33,34], proving useful in promoting fine equiaxed grains in welding[35,36] and traditional casting processes[37,38]. However, successful suppression of the columnar grain structures during AM by ultrasound has not been reported to date.

Based on our long-term studies of ultrasonic grain refinement of light alloys[37,39–42], we employ high-intensity ultrasound to control the solidification and grain structure of AM-fabricated Ti-6Al-4V. This development enables complete transition of columnar prior-β grains into equiaxed fine grains (~100 μm), leading to a 12% improvement in both the yield stress and tensile strength. We further demonstrate that the proposed approach applies to AM of nickel-based superalloy Inconel 625, which shows strong columnar grains as well by fusion-based AM[43–45], and therefore anticipate that it can equally apply to the AM of other metallic materials. Assessment of the ultrasonic field during AM reveals that the selection of the ultrasound transducer element can be an important practical consideration for ultrasonic grain refinement during large-volume AM and a solution is recommended.

## Results

**High-intensity ultrasound during AM of titanium alloy Ti-6Al-4V.** Ti-6Al-4V samples without and with high-intensity ultrasound were prepared using laser-based directed energy deposition (DED). The experimental details are described in the 'Methods' section. The ultrasound was introduced into the melt by directly depositing the alloy on the working surface of the Ti-6Al-4V sonotrode vibrated at 20 kHz (Fig. 1), where the sonotrode material is chosen to be Ti-6Al-4V (for AM of another

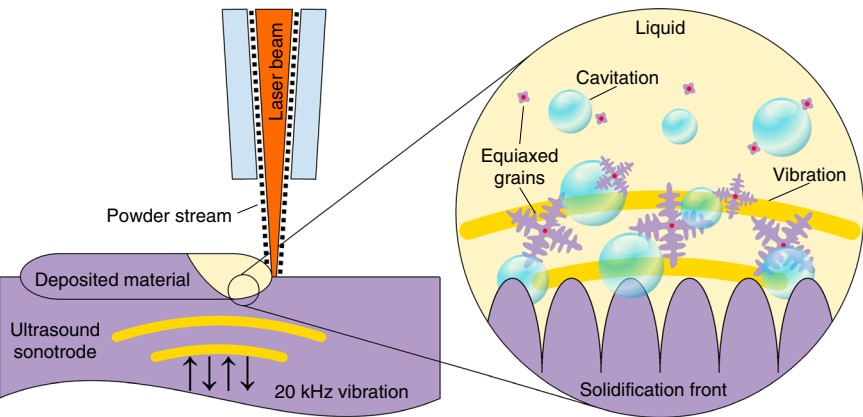

**Fig. 1 High-intensity ultrasound during metal AM.** Cross-sectional schematic showing metal AM by laser-based DED onto an ultrasound sonotrode vibrated at 20 kHz. The formation of acoustic cavitation and streaming in the liquid metal by high-intensity ultrasound can vigorously agitate the melt during solidification, thereby promoting significant structural modification or refinement.

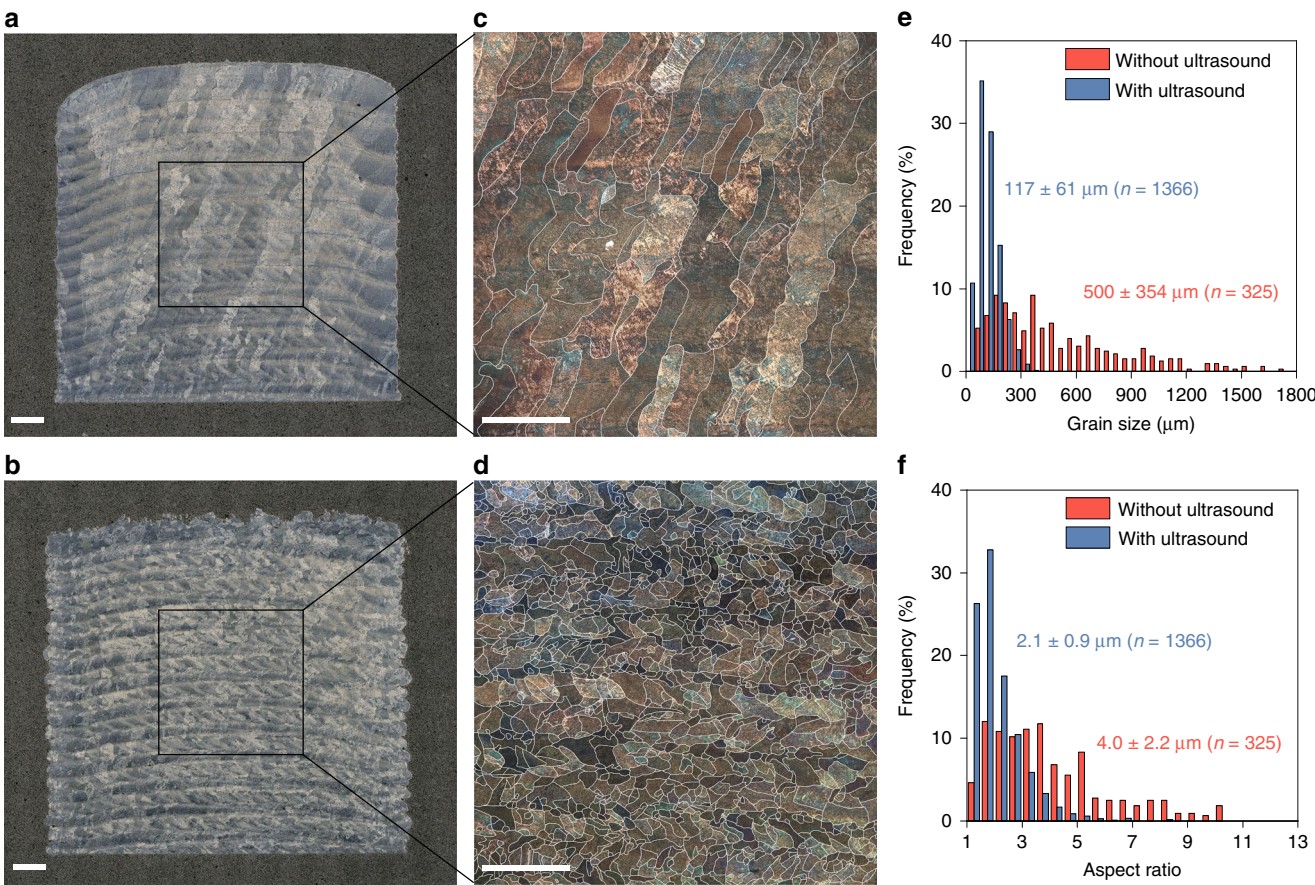

**Fig. 2 Grain refinement of the AM-fabricated Ti-6Al-4V by high-intensity ultrasound. a**, **b** Optical microscopy images of the samples without (**a**) and with (**b**) ultrasound. **c**, **d** Polarized light microscopy images showing large columnar grains (**c**) and fine equiaxed grains (**d**). **e**, **f** Histograms of the prior-β grain size (**e**) and prior-β grain aspect ratio (**f**) for the samples without and with ultrasound measured from traced prior-β grain images (see Supplementary Fig. 1). The prior-β grain boundaries in **c** and **d** are traced in white. Scale bars, 1 mm.

alloy, the sonotrode material can be replaced accordingly). The maximum achievable amplitude at the sonotrode face is 30 μm. The ultrasonic intensity $I$ is defined by[34]:

$$I = \frac{1}{2}\rho c (2\pi f A)^2,\qquad(1)$$

where $\rho$ is the liquid density, $c$ the sound velocity in the liquid, $f$ the frequency and $A$ is the amplitude. The nominal intensity at the sonotrode–melt interface is $>13 \times 10^3$ W cm$^{-2}$ at $A = 30$ μm, where $\rho = 4208$ kg m$^{-3}$ (ref. [46]) and $c = 4407$ m s$^{-1}$ (ref. [47]) for molten Ti. This value is two orders of magnitude greater than the threshold required for cavitation ($I_c$) in molten light metals (~100 W cm$^{-2}$ (ref. [34])), suggesting that the ultrasound applied has the potential to produce significant structural refinement[33,34].

**Microstructure**. Microstructural analysis reveals a substantial difference between the AM-fabricated Ti-6Al-4V samples with and without ultrasound (Fig. 2). The sample without ultrasound exhibits columnar prior-β grains of several millimeters in length and ~0.5 mm in width traversing multiple deposited layers as expected (Fig. 2a, c). In contrast, the sample with ultrasound shows fine (~100 μm), equiaxed prior-β grains (Fig. 2b, d). The effect of ultrasound on grain refinement can be evaluated by examining the change in the prior-β grain number density, which is directly correlated to nucleation[48]. The prior-β grain number density increases from 3.3 mm$^{-2}$ to 65.0 mm$^{-2}$ by ultrasound, confirming that ultrasound enhances nucleation during solidification. The distribution of both the prior-β grain size and prior-β grain aspect

ratio changes dramatically by ultrasound (Fig. 2e, f), reflecting the much-improved prior-β grain structure homogeneity.

To further identify the effect of high-intensity ultrasound on the AM-fabricated Ti-6Al-4V microstructure, the samples with and without ultrasound were analyzed by scanning electron microscopy (SEM). A basketweave-like α–β microstructure is observed inside the prior-β grains in both cases (Fig. 3a–d). The α-lath thickness is similar along the build height with and without ultrasound (see Supplementary Fig. 2). Additionally, no statistical difference is identified in the distribution of the α-lath thickness between the samples without and with ultrasound (Fig. 3e, f). This suggests that the thermal conditions during the β → α transformation that control the scale of the α–β microstructure are largely unaffected by ultrasound. This is not surprising as ultrasound, at the intensity level applied, is not expected to affect the solid-state transformation in metallic alloys.

To assess any potential change in crystallographic texture, electron backscatter diffraction (EBSD) analysis was applied to Ti-6Al-4V samples additively manufactured with and without ultrasound. The results are summarized in Fig. 4. Without ultrasound, the α phase exhibits a clear crystallographic orientation with a maximum multiples of uniform distribution (MUD) value of 4.5 (a measure of the crystallographic preferred orientation strength; maximum MUD = 1.0 corresponds to a random texture). More specifically, without ultrasound, the c-axis of many of the α crystals is tilted ~45° about the columnar or growth direction of the β phase (Fig. 4a, e), measured from the pole figure. This texture has been reported previously[8,11–15]. With

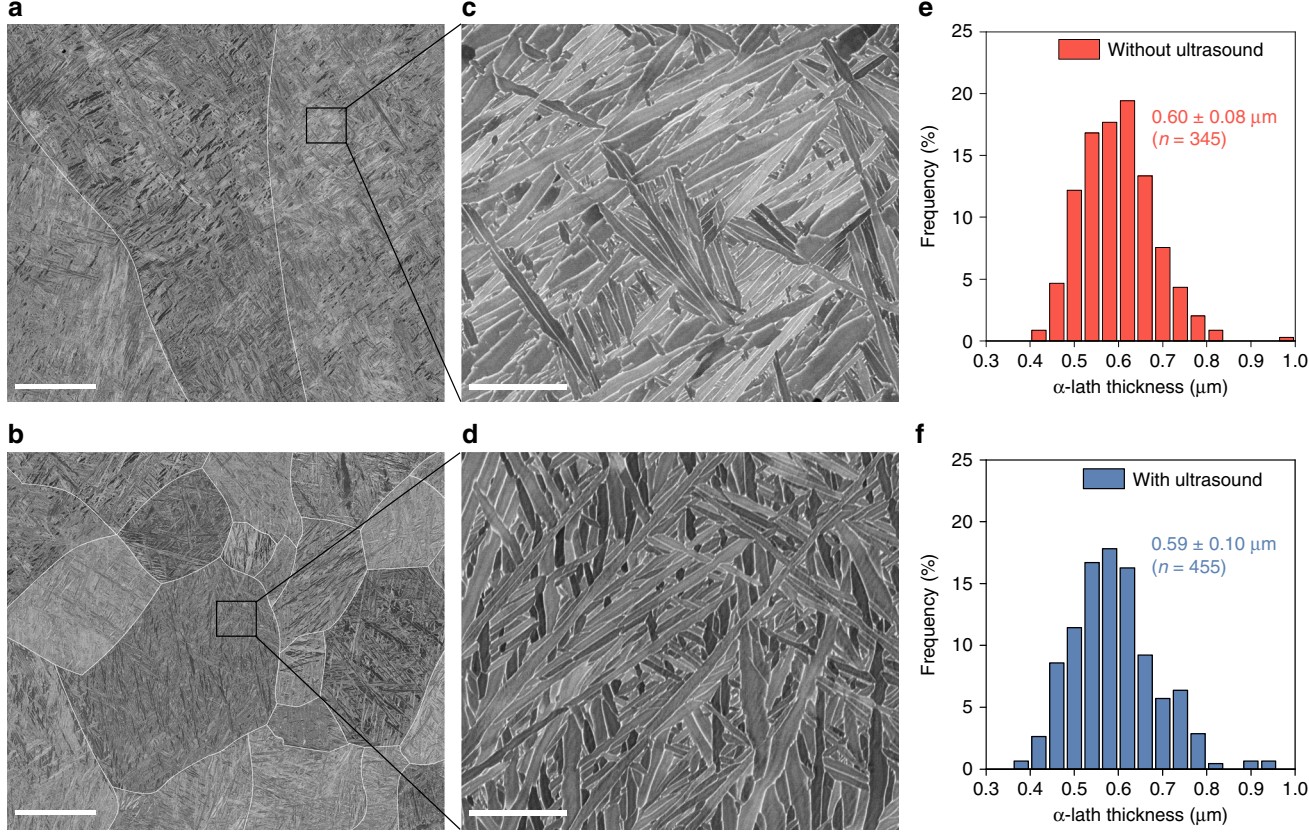

**Fig. 3 Microstructure characterization of the AM-fabricated Ti-6Al-4V with and without high-intensity ultrasound. a–d** SEM images showing the α–β structure inside the prior-β grains of the samples without (**a**, **c**) and with (**b**, **d**) ultrasound. **e**, **f** Histograms of the α-lath thickness of the samples without (**e**) and with (**f**) ultrasound. The prior-β grain boundaries in **a** and **b** are traced in white. Scale bars, 50 μm in **a**, **b** and 5 μm in **c**, **d**.

ultrasound, the maximum MUD value is reduced from 4.5 to 2.0 (Fig. 4c, f), substantially weakening the texture of the α phase.

In the case of the prior-β grains, without ultrasound, the majority of the prior-β grains analyzed show a strong <001> crystallographic orientation (maximum MUD = 6.0; Fig. 4b, g), consistent with previous studies[8,11–15]. With ultrasound, the maximum MUD value for the prior-β grains is reduced from 6.0 to 2.7 (Fig. 4d, h), and the resulting equiaxed prior-β grain structure (Fig. 4d) has effectively avoided the characteristic <001> texture, while no other preferred texture is detected. These observations are consistent with the understanding that an equiaxed grain structure has no preferred crystallographic texture.

**Tensile properties**. Tensile engineering stress–strain curves (Fig. 5a and Supplementary Fig. 3) show that the yield stress $\sigma_y$ and tensile strength $\sigma_{TS}$ of the as-built Ti-6Al-4V are both increased by ~12% by ultrasound (e.g., from 980 ± 13 MPa to 1094 ± 18 MPa for $\sigma_y$, detailed tensile properties are listed in Supplementary Table 1). It should be stressed that the enhanced tensile properties are achieved without increasing the impurity levels of the alloy (see Supplementary Table 2). Both groups of samples show a strain-to-failure ($\varepsilon$) value of ~5%, which is typical of as-built DED-processed Ti-6Al-4V[49,50]. Small pores are found on the fracture surfaces (see Supplementary Fig. 4a, b). Additionally, lack-of-fusion defects perpendicular to the build direction are visible on the polished cross-sections (see Supplementary Fig. 4c, d). The presence of such defects deteriorates the tensile ductility of DED-processed Ti-6Al-4V[49,50].

To put the strength improvement by ultrasound into context, the change in yield stress of AM-fabricated Ti-6Al-4V by ultrasound vs. that by chemical approaches is plotted in Fig. 5b

and also listed in Supplementary Table 3. Deploying ultrasound, without modifying alloy composition, results in a greater increase in yield stress than alloying with B[51], LaB$_6$ (ref. [51]) and C[52].

A recent study has revealed that texture can affect the tensile yield stress of additively manufactured α–β Ti-6Al-4V by 3–5%[15]. This is less than half of the percentage increase in the yield stress observed in Fig. 5a (~12%). To understand the major contributing factor to this increase, Fig. 5c plots the literature data[9,10,14,19,20,23,53–57] and our experimental data on the yield stress of AM-fabricated α–β Ti-6Al-4V vs. the inverse square root of the prior-β grain size (d) (see Supplementary Table 4 for the detailed data). An approximate Hall–Petch relationship is observed. This implies that the resulting equiaxed prior-β grain size has played a major role in improving the yield stress in this study (~7% out of the total 12% of increase).

We note that post-AM heat treatments below the β-transus temperature including hot isostatic pressing are often applied to Ti-6Al-4V for improved strength–ductility combinations and property consistency[8,9]. Such heat treatments do not change the prior-β grain structures[7,10,14]. Hence, the effect of the ultrasound-induced microstructural changes, i.e., the equiaxed prior-β grains, reduced prior-β grain size and substantially weakened texture, on mechanical properties shown in Fig. 5, is expected to survive after common post-AM heat treatments.

**Extension to further alloy systems**. To test the generality of our approach, we have similarly applied high-intensity ultrasound to AM of Inconel 625 using a custom-made stainless-steel 4140 sonotrode (see details in the 'Methods' section). The sample fabricated without ultrasound exhibits columnar primary γ grains of 500 μm in length and 150 μm in width with a strong <001>

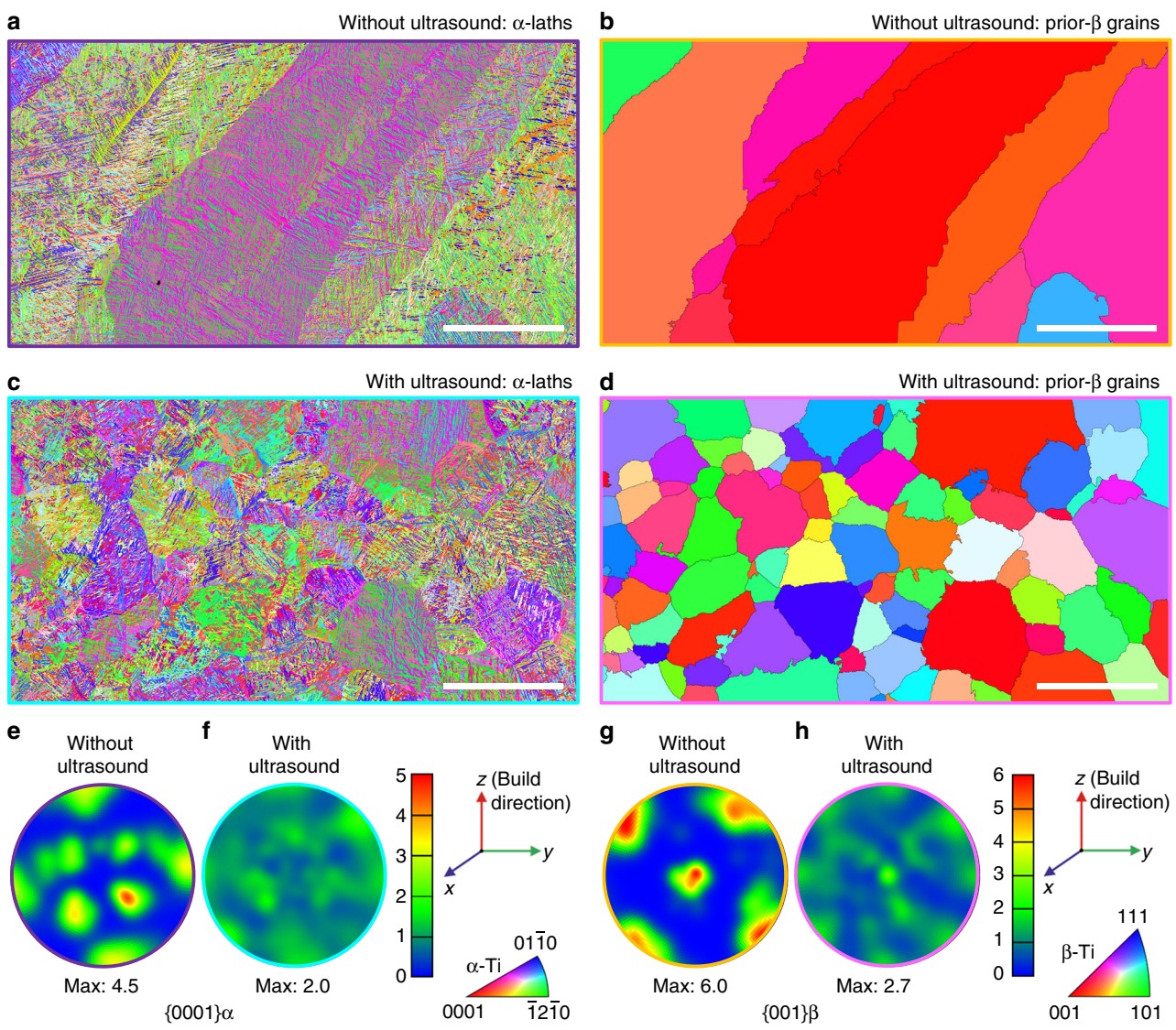

**Fig. 4 Texture changes in AM-fabricated Ti-6Al-4V by high-intensity ultrasound. a, c** Inverse pole figure maps along the build direction (z) for the α phase (measured by EBSD) in samples without (**a**) and with (**c**) ultrasound. **b, d** Inverse pole figure maps along the build direction (z) for the β phase (reconstructed from the α phase maps in **a** and **c**) in samples without (**b**) and with (**d**) ultrasound. **e, f** {0001} contoured pole figures (in MUD: multiples of uniform distribution) of the measured α phase in samples without (**e**) and with (**f**) ultrasound. **g, h** {001} contoured pole figures (in MUD) of the reconstructed β phase in samples without (**g**) and with (**h**) ultrasound. Black lines in **b** and **d** indicate high angle grain boundaries (misorientation >10°). Scale bars, 250 μm.

texture (Fig. 6a, c and Supplementary Fig. 5). In contrast, the application of ultrasound produces predominately equiaxed primary γ grains of only a few microns in size (much finer than Ti-6Al-4V) with a near random crystallographic texture (Fig. 6b, d). This confirms the generality of the ultrasonic approach for AM of different metallic materials.

To further showcase the capability of our approach for solidification control during AM, we fabricated a microstructurally graded Inconel 625 sample that exhibits an alternating columnar/equiaxed/columnar grain structure along its build height, as shown in Fig. 6e. This was achieved by simply turning on and off the high-intensity ultrasound during AM. The approach thus also offers an alternative means of fabricating graded grain structures during AM.

## Discussion

The high-intensity ultrasonic field during AM of Ti-6Al-4V is analyzed as a function of build height to gain fundamental

insights into the potential practical application of ultrasonic grain refinement of alloys made by AM. The sonotrode working face (made of Ti-6Al-4V) is in resonance at the frequency of the piezoelectric transducer when the initial sonotrode length $z_0$ is maintained (Fig. 7a). The effective length of the Ti-6Al-4V sonotrode increases by the sample build height $z$ during AM, where the sample being built is also Ti-6Al-4V by design (the same concept can apply to the AM of other alloys). Consequently, the AM-fabricated part oscillates with different amplitude $A$ values along the sample build height $z$ as a result of changing resonance conditions. The amplitude $A$ as a function of the build height $z$ can be described by a wave equation:

$$A(z) = A_0 \cos \frac{2\pi f}{\nu} z, \tag{2}$$

where $A_0$ is the amplitude at the sonotrode working face (30 μm) and $\nu$ is the sound velocity in solid Ti-6Al-4V (~0.2–0.25 inch μs⁻¹ or ~5000–6100 m s⁻¹ (refs. [58,59])).

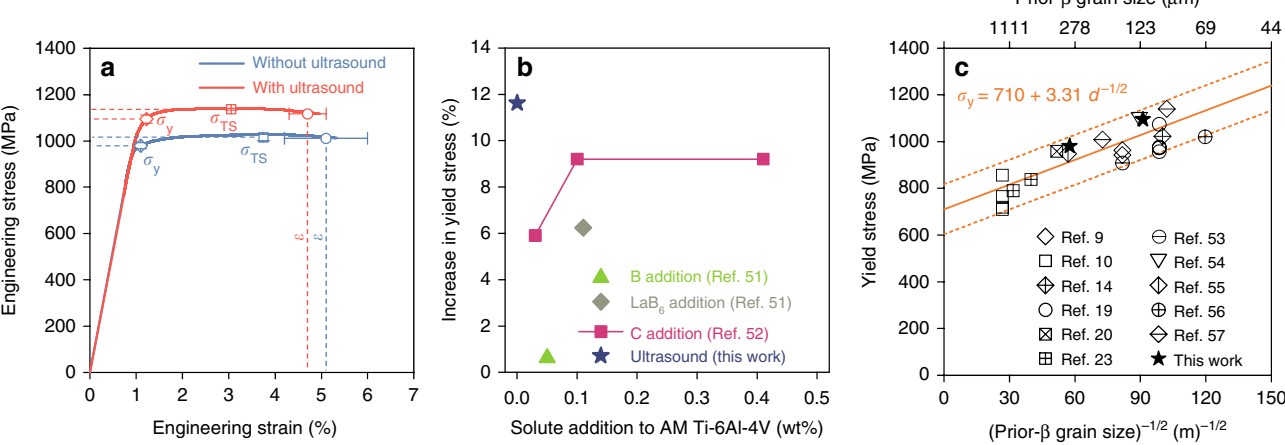

**Fig. 5 Tensile properties of AM-fabricated Ti-6Al-4V. a** Engineering stress–strain curves of the as-built samples without and with ultrasound (all curves are shown in Supplementary Fig. 3). The error bars represent one standard deviation of three tests. **b** Change in yield stress of AM-fabricated Ti-6Al-4V by chemical addition[51,52] compared to ultrasound in this work. See Supplementary Table 3 for data and references. **c** Tensile yield stress with the inverse of square root of prior-β grain size from the literature[9,10,14,19,20,23,53–57] and this work. See Supplementary Table 4 for data and references. The solid line in **c** represents the Hall-Petch line ($\sigma_y = \sigma_0 + kd^{-1/2}$, $\sigma_0$: friction stress; $k$: material constant; $d$: grain size) of best fit while the dashed lines define $\pm 0.15\sigma_0$ (where $\sigma_0 = 710$ MPa) along the linear fit.

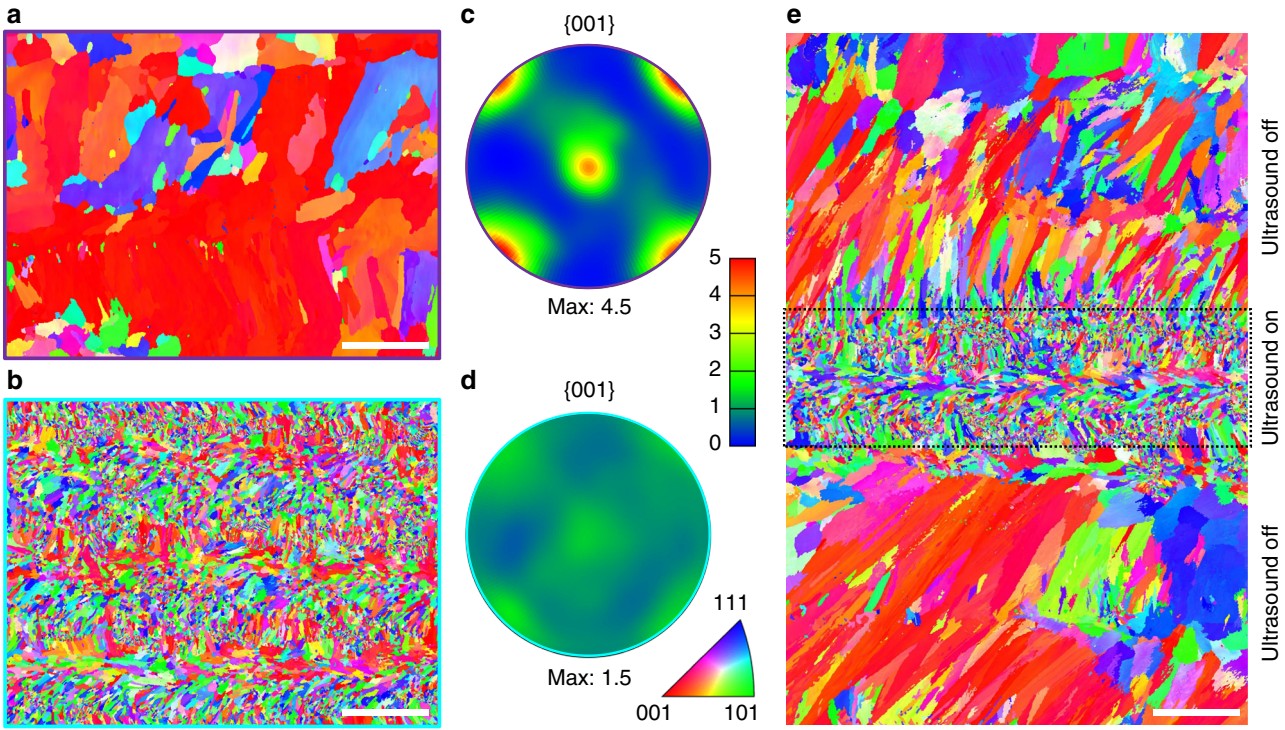

**Fig. 6 AM of Inconel 625 with and without high-intensity ultrasound. a, b** Inverse pole figure maps along the build direction ($z$) for the γ phase in samples without (**a**) and with (**b**) ultrasound. **c, d** {001} contoured pole figures (in MUD: multiples of uniform distribution) of the γ phase in samples without (**c**) and with (**d**) ultrasound. The contoured pole figure in **c** was obtained using the larger area EBSD data provided in Supplementary Fig. 5. **e** Inverse pole figure map along the build direction ($z$) of a sample fabricated by turning the ultrasound on and off during AM. Scale bars, 250 μm.

By combining Eqs. (1) and (2), the ultrasound intensity $I$ as a function of the build height $z$ is given by:

$$I(z) = \frac{1}{2}\rho c \left(2\pi f A_0 \cos\frac{2\pi f}{v}z\right)^2. \quad (3)$$

The intensity ($I$) values calculated from Eq. (3) up to a build height of 500 mm are plotted in Fig. 7b, which vary in a wave-like pattern with a periodicity of ~125 mm. The intensity applied during AM of a 10-mm high Ti-6Al-4V sample remains effectively constant and is two orders of magnitude greater than the threshold for acoustic cavitation, which is essential for significant ultrasonic grain refinement[33,34]. These results can therefore be used to explain the experimental observations of the 10-mm high Ti-6Al-4V sample with high-intensity ultra-sound, which shows fine equiaxed prior-β grains along its height.

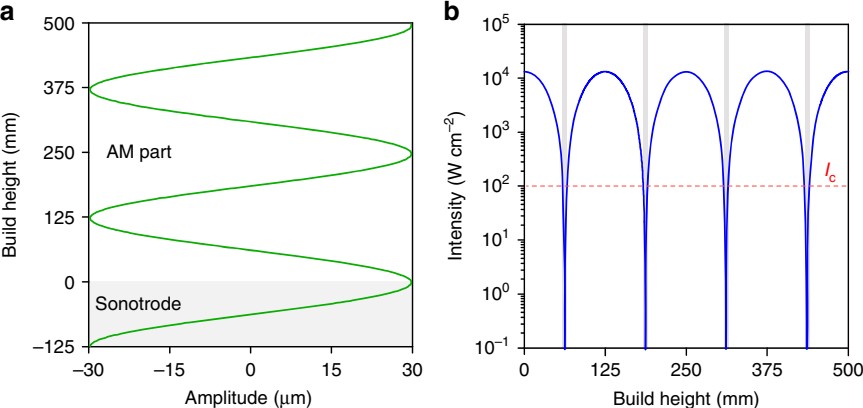

**Fig. 7 High-intensity ultrasound conditions during AM of Ti-6Al-4V. a** Change in amplitude by Eq. (2) versus the vertical axis of the acoustic system. **b** Change in intensity by Eq. (3) versus build height. The red dashed line in **b** corresponds to the intensity required to overcome the cavitation threshold in molten light metals ($I_c \geq 100\ \mathrm{W\,cm^{-2}}$ (ref. [34])). The gray regions in **b** denote when cavitation is non-operative.

The ultrasound intensity ($I$) initially drops from the peak value to zero when the build height increases from zero to ~62.5 mm (the acoustic half wavelength, Fig. 7b) and then returns to the peak value when the build height reaches the acoustic wavelength (~125 mm). This pattern repeats itself when the build height increases further (Fig. 7b). Consequently, the ultrasound intensity $I$ required to produce cavitation for structural refinement is intermittently unsatisfied. This reveals the limitation of using a piezoelectric transducer, which is unable to maintain a constant amplitude $A$ with sample build height $z$. To overcome this problem, it is better to use a magnetostrictive transducer that can be automatically tuned to the variable resonance condition by adjusting the frequency. This, coupled to a specially designed slotted block sonotrode with a wide output face, would enable the peak intensity to be maintained with build height and facilitate grain refinement throughout large AM-fabricated parts. However, it should be noted that the geometry of the additively manufactured components may affect the ultrasonication conditions, which could become a practical concern when fabricating complex shapes and deserves further investigation.

For conventional ultrasonic grain refinement in a large volume of melt, the basic grain refinement mechanisms are generally clear, i.e., cavitation is essential for the production of a large number of nuclei or crystallites (up to four different mechanisms may be operative[34,40])[31,60–62], while acoustic streaming is important for their distribution from the sonotrode region to the rest of the melt[63,64]. Due to the very small melt pool (~0.8 mm) during AM, the distribution effect of acoustic streaming on grain refinement can be assumed to be minimal, because the entire small melt pool is effectively ultrasonicated. As detailed in Supplementary Note 1, the DED process employed in this study offers far more than sufficient time for cavitation to occur (~0.00003 s) in the melt pool. On the other hand, the ultrasound intensity employed in this study far exceeds the threshold for cavitation in molten Ti-6Al-4V, estimated according to the surface tension of molten Ti-6Al-4V (~1.05 N m$^{-1}$) during AM, compared with that of molten Al (~0.9 N m$^{-1}$) and the detailed measurements of cavitation in molten Al by Eskin[34]. In that regard, cavitation can be assumed to be the predominant reason for the grain refinement observed.

Finally, this work is restricted to DED for ultrasonic grain refinement during AM. Previous studies have shown that stimulating solidification control during wire-fed welding processes is possible by ultrasonically vibrating the weld pool[35,65]. Since both wire-fed welding and wire-fed AM deposition processes share similar fundamental principles, we anticipate that the scheme presented in this study can be extended to wire-fed AM processes. However, the vibrating sonotrode may risk disrupting the layer of powder after recoating on a powder bed fusion AM system. In that regard, the inoculation path for grain refinement may be more applicable to metal AM by powder bed fusion processes.

To conclude, high-intensity ultrasound has been used to address a long-standing problem in metal AM, namely strong epitaxial growth facilitating the formation of columnar grains oriented along the build direction. Herein, the application of ultrasound during AM of Ti-6Al-4V enables the formation of a fully equiaxed structure, which improves the microstructural homogeneity, significantly reduces the prior-β grain size and substantially weakens the solidification texture. This work highlights the important role of prior-β grain refinement in the tensile properties of AM Ti-6Al-4V. Assessment of the ultrasonic conditions reveals that the selection of the ultrasound transducer element can be an important practical consideration for structural refinement of large-volume AM-fabricated parts and the use of a magnetostrictive transducer is recommended. To assess the generality of our approach, the ultrasonic grain refinement method is successfully applied to the AM of Inconel 625, including the creation of an alternating columnar/equiaxed/columnar Inconel 625 grain structure along the build height by simply switching on and off the ultrasound during AM. We expect that this technique can be extended to the AM of other metallic materials.

## Methods

**Sample preparation**. Gas-atomized extra low interstitial (ELI) grade Ti-6Al-4V powder of 45–90 μm was used for AM by a laser-based DED system (Trumpf, TruLaser Cell 7020). Samples consisted of 10 mm × 10 mm × 10 mm cubes for microstructural examination and 24 mm × 8 mm × 10 mm (length, width and height) blocks for tensile testing. The samples without high-intensity ultrasound were built on a Ti-6Al-4V plate using a laser power of 250 W, laser spot size of 0.61 mm, scan speed of 600 mm min$^{-1}$ and overlap ratio of 50%. The ultrasound provides additional input power to the melt in the form of acoustic power. To prevent overheating of the melt pool, the laser power was reduced from 250 W to 150 W by keeping other parameters unchanged. The samples were built on the working face of a 25-mm diameter Ti-6Al-4V sonotrode (Fig. 1). Detailed optical microscopy analysis has revealed that the porosity on the polished cross-sections of the Ti-6Al-4V samples with and without ultrasound was similar, in the range of 0.7–0.9 area% (see Supplementary Fig. 6). The sonotrode working face was driven by a 500-W piezoelectric transducer (Sonic Systems, L500) operating at 20 kHz.

Gas-atomized Inconel 625 powder of 45–90 μm was used for AM of Inconel 625 by laser-based DED (Trumpf, TruLaser Cell 7020). Cuboidal samples with dimensions of 10 mm × 10 mm × 5 mm (length, width, height) were built for microstructural characterization. The sample without high-intensity ultrasound was built on a 4140 stainless-steel plate using a laser power of 300 W, laser spot size

of 0.61 mm, scan speed of 600 mm min$^{-1}$ and overlap ratio of 50%. The sample with ultrasound was built on a 25-mm diameter 4140 stainless-steel sonotrode using the same parameters for the sample without ultrasound, other than a reduced laser power of 120 W.

Chemical analysis using inductively coupled plasma-atomic emission spectroscopy (ICP-AES) and LECO combustion was done on the Ti-6Al-4V samples with and without high-intensity ultrasound to evaluate if any extra interstitial solute was introduced into the melt during ultrasonic irradiation. This is because an increase in interstitial content may account for both grain refinement due to constitutional supercooling[66] and increased strength through solid solution strengthening[67]. The results indicate that there was negligible additional interstitial pickup by ultrasound (see Supplementary Table 2).

**Microstructure characterization and tensile testing.** The Ti-6Al-4V cube samples were cut in half along the build direction and prepared for microstructural characterization by standard techniques with final polishing by 0.04 μm colloidal silica suspension. The microstructure was first examined by SEM (FEI, Verios 460L) in the backscattered electron mode. Then the samples were etched with Kroll's reagent and examined by optical microscopy under polarized light to distinguish the prior-β grains by their crystal orientation[68]. The prior-β grain boundaries were manually traced and the total number of prior-β grains within each view was divided by the area to obtain the prior-β grain number density (mm$^{-2}$). The images of the traced prior-β grains (see Supplementary Fig. 1c, d) were analyzed using ImageJ software[69] to obtain statistical distributions of the prior-β grain size and prior-β grain aspect ratio. Lamellar spacing measurements were made on SEM images taken along the build height of each sample by the linear intercept method. Three fields of view were analyzed at each build height resulting in ~40 line segments per data point. EBSD analysis (accelerating voltage of 20 kV, probe current of 13 nA, step size of 0.5 μm, working distance of 15 mm and sample-tilt angle of 70°) was conducted using a scanning electron microscope (JEOL, JSM-7200F) equipped with an EBSD detector (Oxford Instruments, NordlysMax$^2$). A montage of 56 individual tiles of EBSD data comprising a total area of ~0.6 mm × 1.2 mm per sample was used to obtain the texture data. The orientation information of the β phase was reconstructed from the α phase EBSD data using the software package ARPGE[70].

The Inconel 625 samples were sectioned along the build direction and prepared for microstructural characterization by standard techniques with final polishing by 0.04 μm colloidal silica suspension. Microstructural analysis was conducted by EBSD (accelerating voltage of 20 kV, probe current of 16 nA, step size of 1.5 μm (Fig. 6a and Supplementary Fig. 5) or 0.5 μm (Fig. 6b, e), working distance of 15 mm and sample-tilt angle of 70°) using a scanning electron microscope (JEOL, JSM-7200F) equipped with an EBSD detector (Oxford Instruments, NordlysMax$^2$).

The as-built Ti-6Al-4V block samples were cut into flat tensile specimens transverse to the build direction with a gauge length of 12 mm, width of 2 mm and thickness of 1 mm. Tensile testing with an initial strain rate of $2.5 \times 10^{-4} \, \text{s}^{-1}$ was performed on three specimens per condition using a universal testing machine (MTS, 810) equipped with a non-contact laser extensometer (MTS, LX500).

## Data availability

The data that support the findings of this study are available from the corresponding author upon reasonable request.

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

## Acknowledgements

This research work was supported by the Australian Research Council (ARC) Discovery Projects DP150104719 and DP140100702 and the ExoMet Project co-funded by the European Commission's 7th Framework Programme (contract FP7-NMP3-LA-2012-280421), by the European Space Agency and by the individual partner organizations. M.A.E., D.Q. and D.H.S. further acknowledge the support of the ARC Discovery Project DP160100560. M.J.B. acknowledges the support of the School of Mechanical and Mining Engineering at The University of Queensland as well as the ARC Discovery Early Career Researcher Award DE160100260. We thank both the Microscopy and Microanalysis Facility (RMMF) and the Advanced Manufacturing Precinct (AMP) at RMIT University for their facilities and technical assistance. In addition, we thank Sonic Systems Ltd (Somerset, UK) for their timely design and support.

## Author contributions

C.J.T. and M.Q. conceived the idea. C.J.T., D.Q. and D.Z. fabricated the samples. C.J.T. and E.W.L. performed the tensile tests. C.J.T. performed all other remaining experiments. C.J.T. drafted the manuscript, and C.J.T., M.A.E., D.Q., D.Z., M.J.B., E.W.L., M.B., D.H.S. and M.Q. interpreted, discussed and edited the manuscript. C.J.T. and M.Q. finalized the manuscript, including preparing the detailed response letter. M.Q. supervised the work.

## Competing interests

The authors declare no competing interests.
