## [Transparent Peer Review File · Nature Communications]

Reviewers' comments:

Reviewer #1 (Remarks to the Author):

The authors describe methods to control solidification during additive manufacturing, the influence of the application of high intensity ultrasound on the resulting microstructure, and the resulting effects of microstructure on the mechanical properties.

The authors are well respected, especially in the area of solidification control, and have previously demonstrated the application of ultrasound to disrupt epitaxial solidification during additive manufacturing. I was quite pleased to have the opportunity to review this paper, but have mixed assessments of the findings. To differentiate the papers strengths and less-well developed claims, I will break this review into: process, microstructure, properties. I expect this breakdown may be helpful for the authors as they consider revisions.

Regarding the process, the technique is exciting. The claim that the implosive collapse of bubbles (I might rewrite as cavities, as "bubbles" may imply containing a gas, which is not the case) results in such dramatic increases in pressures and temperatures is nicely presented, and novel to additive manufacturing. This is the greatest strength of the paper.

Regarding microstructure, the authors miss the opportunity to demonstrate the most significant microstructures impact realized by this processing method, that is, the disruption of the 001 bcc beta grain texture. The authors appropriately refer to this phenomenon, but do not measure the texture in the material processed with the ultrasound applied. This is a significant weakness of the paper, and sets the authors up to make claims that are not founded in their discussion of the properties. I strongly recommend that the authors analyze the crystallographic orientation of the alpha phase and thereby deduce any change in texture of the parent beta grains.

Regarding properties, the authors (having missed quantifying texture, which has a pronounced influence on mechanical behavior, all else being equal) then proceed to make claims regarding the activation of the Hall-Petch phenomenon as the reason for the increase in strength. This is the weakest section of the paper. In alpha+beta titanium alloys, including Ti-64, there is little effect of grain size on the strength, especially when the basketweave Widmanstätten microstructure dominates. This is due to the fact that there is not a single slip system that is active from one grain boundary to another...the presence of basketweave (which is what is present in these microstructures) disrupts slip, and thereby disrupts the traditional dislocation pileup, which would lead to the Hall-Petch effect. The authors invoke an equation from TiAl gamma+alpha₂, which has large "colonies" of single orientations, and can exhibit a Hall-Petch effect. But, for Ti-6Al-4V, an "inverse" Hall-Petch relationship has actually been reported in the literature (see Metallurgical and materials transactions A, 37(3), 559-566, which was attributed to the competition of microstructure type within the transformed grain). In invoking the Hall-Petch effect (which may be a contributor) but in the absence of any quantification of texture, the authors are potentially missing the primary factor. Somewhat recently, other authors (Acta Materialia, 133, 120-133) have shown that texture alone can account for about half of the change in strength that the authors observe.

If possible, I would encourage the authors to obtain EBSD (or equivalent orientation) data. This would not only be helpful to convey the potentially seminal importance of this work (using ultrasound to disrupt texture) and increase its visibility and impact, but also provide clarity to their latter discussion on properties.

Beyond this overall review, I would like to make the following points:

- * property anisotropy may be bad, reducing properties, but conversely, could be good in the hands of an expert. I would suggest editing line 19 to be a bit more circumspect.
- * Lines 54-56...some literature shows that coarse grains actually strengthen Ti-6Al-4V, as it

promotes the formation of basketweave

* Figure 1: Very nice!! (in general, the figures are very well done - my compliments to the authors)

* Line 182: The assumption that B, LaB₆, and C are impurities is not quite right. An impurity means something that should not be there by design, whereas the inclusion of these elements can be intentional. I understand what the authors are saying, but would suggest changing "while without introducing impurity to the alloy" to "without modifying the composition of the alloy".

* Equation 189 is applicable when the adjacent lamellae is of the same orientation and dislocations have a means of easily transferring...so, suitable for TiAl, but probably not for Ti-64

* Lines 193-197 are convoluted, and (if kept) should be re-read.

* Line 219 - are the amplitude values dependent upon the geometry - i.e., I assume a thick part or a corner would behave differently than a thin part. Please comment.

* Line 237 - the "It" is vague. (other than this minor correction, the paper was well-written).

* Line 300 - when reducing the laser power to 150 W, was the expected increase in power due to the ultrasound calculated, or was this value arbitrarily selected?

Reviewer #2 (Remarks to the Author):

The authors here demonstrate a technique whereby an ultrasonic sonotrode embedded in the build tray of a DED printer can be used to create fine, equiaxed grains in a Ti-6-4 part, which imparts up to 12% improvement in the tensile yield strength of the as-cast part. The authors claim that this technique can be widely applied to other alloys and other additive manufacturing techniques.

For starters, let me say that I found the experiment used in this paper to be very clever and the analysis of why the process works is sound. I really enjoyed reading this paper and I think it will likely change the way that future metal 3D printers are built. My further comments are not so much about what is shown in the current paper, but rather what isn't shown. The authors make broad and sweeping claims about the applicability of the process they've developed but they only show it for one DED printer and one alloy system (Ti-6-4)

Starting with titanium: DED is one of really 4 competing AM methods for producing Ti. There is LPBF, EPBF, wire-fed E-beam and DED. Arguably, the EPBF parts made by Arcam printers are the most mature but the authors here only discuss DED. While DED is definitely useful for Ti, there is no discussion nor experiments showing that this sonotrode method would work on the powder-bed printers or the wire-fed printers. To claim broad applicability of the process, even for Ti, I would think that you would have to have a discussion about the applicability of the process to other AM systems. For example, wouldn't the sonotrode disrupt the powder after re-coating on a powder bed system? If not, you should demonstrate it by putting your sonotrode in a PBF printer. If it does, you need to scale back your broad applicability argument and put in some discussion about which types of additive processes can use this technique. The inoculants path may be better for powder bed printers, and this needs to be mentioned.

Next is the mechanical properties of the Ti: You only show tension tests from parts in the as-built conditions, comparing your method with the standard DED printing. However, the vast majority of people using AM titanium alloys are applying HIP and heat treatments to the parts before being used in service. To not show the properties of your parts compared with HIP and heat treated parts seems to be a huge omission. After all, if the properties after HIP and heat treatment are the same between the printing with the sonotrode and printing without it, then you need to change your discussion to only focus on as-printed titanium parts. This makes your process much less desirable if there is no benefit for the vast majority of people who will be heat treating. At the very least, you might want to show parts that have been stress relieved to see what the difference is between the residual stresses using your method or without it.

Next is fatigue. You mention off-hand that the equiaxed Ti parts will have better fatigue, but you don't show this. It would be notable if the fatigue of Ti parts using the sonotrode method is superior to Ti parts made other ways when no heat treatments are applied. Since the parts you show are made through DED, they will have to be machined anyways, due to the low resolution of the DED printer. Will your method produce better properties once the parts are machined?

Next is alloys: You mention that the process can be used for other alloy systems to achieve refined grains. Why not show this in some other alloy to generalize the process? Print some steel and show that the grains are different. Does this process work across alloy systems?

Lastly, the paper seems to be lacking discussion about using this technique to produce parts with microstructural gradings. If the sonotrode can be turned on and off during a print, couldn't you produce a part with both equiaxed and columnar grains? Why didn't you do that? That would have been an amazing demonstration of this technique. Perhaps you should make a few more Ti samples and add this to the paper.

Overall, I thought the paper was very interesting in terms of what was achieved in Ti. However, the process was demonstrated for DED Ti and the discussion claims that it could be used in multiple alloys across multiple printers. I do not think that is supported. The authors need to either add additional details and experiments to justify their claims or they need to clearly point out the limitations of the technique to their experimental setup.

Regards,
Douglas Hofmann

Reviewer #3 (Remarks to the Author):

The article reports an observation that ultrasound promotes the formation of equiaxed microstructure over columnar structure in laser metal AM processes. Ti-6Al-4V is chosen as the model alloy for this study. The authors suggest a hypothesis that cavitation-enhanced nucleation enables the grain refinement. The authors support the conclusion by the grain characterization, tensile strength, and calculation of ultrasound intensity.

The result is interesting and relevant in the area of additive manufacturing. However, to be publishable in this journal, the paper needs to be significantly improved. The following are some areas for improvement:

1. A direct proof of the hypothesis would be necessary. The paper is still at the stage of hypothesis formulation. It is likely that ultrasound could generate cavitations that enhance nucleation. There is no conclusive and direct evidence in the manuscript proving this hypothesis. The calculation of ultrasound intensity helps, but not sufficient to support the conclusion.
2. The authors have previously published works on ultrasound induced grain refinement. What is the new physics when applying this technique in laser fusion AM process? Does the high G in AM process change any of the characteristics comparing with previous studies? It is not clear how this paper is different from previous published works in terms of grain refinement mechanism.
3. Any effects of the ultrasound on the deposited powders? Does it induce movement of powders? Any effects on the final porosity comparing with that without ultrasound?

Response Letter

We greatly appreciate the insightful comments received from all the reviewers and are pleased to show that we have (i) quantified the crystallographic texture with and without ultrasound; (ii) successfully demonstrated this technique on Inconel 625 using custom-made new sonotrodes; (iii) demonstrated the “amazing” ultrasound on-and-off effect on grain structure; (iv) re-discussed the Hall-Petch relationship for Ti-6Al-4V, and (v) analyzed the cavitation effect based on ultrafast in situ synchrotron X-ray imaging results of cavitation in molten metals from the literature. We hope we have reasonably addressed all the major concerns. Our response to each comment is detailed below.

Reviewer #1:

Comment #1: Regarding microstructure, the authors miss the opportunity to demonstrate the most significant microstructures impact realized by this processing method, that is, the disruption of the 001 bcc beta grain texture. The authors appropriately refer to this phenomenon, but do not measure the texture in the material processed with the ultrasound applied. This is a significant weakness of the paper, and sets the authors up to make claims that are not founded in their discussion of the properties. I strongly recommend that the authors analyze the crystallographic orientation of the alpha phase and thereby deduce any change in texture of the parent beta grains.

Response: Thank you. We have performed detailed electron backscatter diffraction (EBSD) analyses. The results are summarized in Fig. 4. For each sample, we attained 56 individual maps of EBSD data for the α phase (step size of 0.5 μm), comprising a total area of $\sim 0.6 \text{ mm} \times 1.2 \text{ mm}$. We then reconstructed the β phase orientation information from the EBSD data obtained from the α phase using the software package *ARPGE* [1]. On this basis, we are able to confirm that the application of ultrasound has disrupted the strong $\langle 001 \rangle$ orientation texture of the β phase. We have added the following text to the revised manuscript:

“To assess any potential change in crystallographic texture, electron backscatter diffraction (EBSD) analysis was applied to Ti-6Al-4V samples additively manufactured with and without ultrasound. The results are summarized in Fig. 4. Without ultrasound, the α phase exhibits a clear crystallographic orientation with a maximum multiples of uniform distribution (MUD) value of 4.5 (a measure of the crystallographic preferred orientation strength; maximum MUD = 1.0 corresponds to a random texture). More specifically, without ultrasound, the c -axis of

many of the α crystals is tilted $\sim 45^\circ$ about the columnar or growth direction of the β phase (Fig. 4a, e), measured from the pole figure. This texture has been reported previously^{8,14-18}. With ultrasound, the maximum MUD value is reduced from 4.5 to 2.0 (Fig. 4c, f), substantially weakening the texture of the α phase.”

In the case of the prior- β grains, without ultrasound, the majority of the prior- β grains analyzed show a strong $\langle 001 \rangle$ crystallographic orientation (maximum MUD = 6.0; Fig. 4b, g), consistent with previous studies^{8,14-18}. With ultrasound, the maximum MUD value for the prior- β grains is reduced from 6.0 to 2.7 (Fig. 4d, h), and the resulting equiaxed prior- β grain structure (Fig. 4d) has effectively avoided the characteristic $\langle 001 \rangle$ texture, while no other preferred texture is detected. These observations are consistent with the understanding that an equiaxed grain structure has no preferred crystallographic texture.”

Fig. 4 Texture changes in AM-fabricated Ti-6Al-4V by high-intensity ultrasound. a, c Inverse pole figure maps along the build direction (z) for the α phase (measured by EBSD) in samples a without and c with ultrasound. b, d Inverse pole figure maps along the build direction (z) for the β phase (reconstructed from the α phase maps in a and c) in samples b without and d with ultrasound. e, f $\{0001\}$ contoured pole figures of the measured α phase in samples e without and f with ultrasound. g, h $\{001\}$ contoured pole figures of the reconstructed β phase in samples g without and h with ultrasound. Black lines in b and d indicate high angle grain boundaries (misorientation $>10^\circ$). Scale bars, 250 μm .

Comment #2: Regarding properties, the authors (having missed quantifying texture, which has a pronounced influence on mechanical behavior, all else being equal) then proceed to make claims regarding the activation of the Hall-Petch phenomenon as the reason for the increase in strength. This is the weakest section of the paper. In alpha+beta titanium alloys, including Ti-64, there is little effect of grain size on the strength, especially when the basketweave Widmanstätten microstructure dominates. This is due to the fact that there is not a single slip

system that is active from one grain boundary to another...the presence of basketweave (which is what is present in these microstructures) disrupts slip, and thereby disrupts the traditional dislocation pileup, which would lead to the Hall-Petch effect. The authors invoke an equation from TiAl $\gamma+\alpha_2$, which has large "colonies" of single orientations, and can exhibit a Hall-Petch effect. But, for Ti-6Al-4V, an "inverse" Hall-Petch relationship has actually been reported in the literature (see Metallurgical and materials transactions A, 37(3), 559-566, which was attributed to the competition of microstructure type within the transformed grain). In invoking the Hall-Petch effect (which may be a contributor) but in the absence of any quantification of texture, the authors are potentially missing the primary factor. Somewhat recently, other authors (Acta Materialia, 133, 120-133) have shown that texture alone can account for about half of the change in strength that the authors observe.

If possible, I would encourage the authors to obtain EBSD (or equivalent orientation) data. This would not only be helpful to convey the potentially seminal importance of this work (using ultrasound to disrupt texture) and increase its visibility and impact, but also provide clarity to their latter discussion on properties.

Response: Thank you. Following this suggestion, as shown earlier, we have carried out detailed texture analyses. Indeed, there is a distinct difference in texture with and without ultrasound. Therefore, we have added the following text to the discussion. Please also refer to our response to your comment #6, which further addresses your concern about the Hall-Petch relationship.

“A recent study has revealed that texture can affect the tensile yield stress of additively manufactured α - β Ti-6Al-4V by 3-5%¹⁸. This is less than half of the percentage increase in the yield stress observed in Fig. 5a (~12%). To understand the major contributing factor to this increase, Fig. 5c plots the literature data^{9,10,17,22,23,26,53-57} and our experimental data on the yield stress of AM-fabricated α - β Ti-6Al-4V vs. the inverse square root of the prior- β grain size (d) (see Supplementary Table 4 for the detailed data). An approximate Hall-Petch relationship is observed. This implies that the resulting equiaxed prior- β grain size has played a major role in improving the yield stress in this study (~7% out of the total 12% of increase).”

Fig. 5c Tensile yield stress of AM-fabricated Ti-6Al-4V with the inverse of square root of prior- β grain size from the literature^{9,10,17,22,23,26,53-57} and this work. See Supplementary Table 4 for data and references. The solid line in represents the line of best fit while the dashed lines define $\pm 0.15\sigma_0$ along the linear fit.

Comment #3: Property anisotropy may be bad, reducing properties, but conversely, could be good in the hands of an expert. I would suggest editing line 19 to be a bit more circumspect.

Response: We agree and have made the following change:

“These long columnar grains can cause property anisotropy, which can be an important concern for component qualification or targeted application.”

Comment #4: Lines 54-56...some literature shows that coarse grains actually strengthen Ti-6Al-4V, as it promotes the formation of basketweave.

Response: We appreciate this comment and have made the following changes:

“In addition, the coarse columnar prior- β grains may further degrade the strength of Ti-6Al-4V according to the Hall-Petch relationship established for lamellar α - β Ti-6Al-4V²⁴⁻²⁶ (exceptions can exist²⁷).”

Comment #5: Line 182: The assumption that B, LaB₆, and C are impurities is not quite right. An impurity means something that should not be there by design, whereas the inclusion of these elements can be intentional. I understand what the authors are saying, but would suggest changing "while without introducing impurity to the alloy" to "without modifying the composition of the alloy".

Response: We agree, and the suggestion has been followed:

“Deploying ultrasound, without modifying alloy composition, results in a greater increase in yield stress than alloying with B⁵¹, LaB₆⁵¹ and C⁵².”

Comment #6: Equation line 189 is applicable when the adjacent lamellae is of the same orientation and dislocations have a means of easily transferring...so, suitable for TiAl, but probably not for Ti-6Al.

Response: Thank you for this constructive comment. In addition to the reverse Hall-Petch relationship mentioned previously [2], the current literature [3-8] has more commonly shown that the classical Hall-Petch relationship is applicable to α - β Ti-6Al-4V in the form of:

$$\sigma_y = \sigma_0 + kd^{-1/2}$$

where d can be the prior- β grain size [4-5, 7], α -lath thickness (or α lamellar spacing) [4-6, 8], or α - β colony size [3]. The above equation has also been found to be applicable to additively manufactured Ti-6Al-4V with either a basketweave-like microstructure [7] or a mixed microstructure of colonies (of single orientations) and basketweaves [5]. This agrees with our Fig. 5c (a new plot). On this basis, we have re-written this section of the paper (the original equation line has been removed), which is the same response to your comment #2.

Comment #7: * Lines 193-197 are convoluted, and (if kept) should be re-read.

Response: Thank you. Those whole four lines have been eliminated (irrelevant now due to the response to your comment #2).

Comment #8: * Line 219 - are the amplitude values dependent upon the geometry - i.e., I assume a thick part or a corner would behave differently than a thin part. Please comment.

Response: Thank you. We have added the following comments to the revised manuscript (we will need to identify this in the future with different sonotrodes):

“However, it should be noted that the geometry of the additively manufactured components may affect the ultrasonication conditions, which could become a practical concern when fabricating complex shapes and deserves further investigation.”

Comment #9: * Line 237 - the "It" is vague.

Response: Agree and we have replaced the original statement with the following:

“The ultrasound intensity (I) initially drops from the peak value to zero when the build height increases from zero to ~62.5 mm (the acoustic half wavelength, Fig. 7b) and then returns to the peak value when the build height reaches the acoustic wavelength (~125 mm). This pattern repeats itself when the build height increases further (Fig. 7b).”

Comment #10: Line 300 - when reducing the laser power to 150 W, was the expected increase in power due to the ultrasound calculated, or was this value arbitrarily selected?

Response: The laser power is an important parameter in our technique. We first calculated the extra power from ultrasound, which gives ~125 W. Then we compared with experimental studies and finalized that the actual difference is ~100 W for AM of Ti-6Al-4V (for AM of Inconel 625 the difference is ~180 W). We have added the following to the text:

“The ultrasound provides additional input power to the melt in the form of acoustic power. To prevent overheating of the melt pool, the laser power was reduced from 250 W to 150 W by keeping other parameters unchanged. The samples were built on the working face of a 25-mm diameter Ti-6Al-4V sonotrode (Fig. 1).”

Reviewer #2:

Comment #1: Starting with titanium: DED is one of really 4 competing AM methods for producing Ti. There is LPBF, EPBF, wire-fed E-beam and DED. Arguably, the EPBF parts made by Arcam printers are the most mature but the authors here only discuss DED. While DED is definitely useful for Ti, there is no discussion nor experiments showing that this

sonotrode method would work on the powder-bed printers or the wire-fed printers. To claim broad applicability of the process, even for Ti, I would think that you would have to have a discussion about the applicability of the process to other AM systems. For example, wouldn't the sonotrode disrupt the powder after re-coating on a powder bed system? If not, you should demonstrate it by putting your sonotrode in a PBF printer. If it does, you need to scale back your broad applicability argument and put in some discussion about which types of additive processes can use this technique. The inoculants path may be better for powder bed printers, and this needs to be mentioned.

Response: Thank you for raising this important concern. We need to scale back our broad applicability argument. We have amended our manuscript with the following addition:

“Finally, this work is restricted to DED for ultrasonic grain refinement during AM. Previous studies have shown that stimulating solidification control during wire-fed welding processes is possible by ultrasonically vibrating the weld pool^{38,65}. Since both wire-fed welding and wire-fed AM deposition processes share similar fundamental principles, we anticipate that the scheme presented in this study can be extended to wire-fed AM processes. However, the vibrating sonotrode may risk disrupting the layer of powder after recoating on a powder bed fusion AM system. In that regard, the inoculation path for grain refinement may be more applicable to metal AM by powder bed fusion processes.”

Comment#2:

Next is the mechanical properties of the Ti: You only show tension tests from parts in the as-built conditions, comparing your method with the standard DED printing. However, the vast majority of people using AM titanium alloys are applying HIP and heat treatments to the parts before being used in service. To not show the properties of your parts compared with HIP and heat treated parts seems to be a huge omission. After all, if the properties after HIP and heat treatment are the same between the printing with the sonotrode and printing without it, then you need to change your discussion to only focus on as-printed titanium parts. This makes your process much less desirable if there is no benefit for the vast majority of people who will be heat treating. At the very least, you might want to show parts that have been stress relieved to see what the difference is between the residual stresses using your method or without it.

Response: We appreciate these comments and suggestions.

Briefly, post-AM heat treatments of AM-fabricated Ti-6Al-4V are typically applied below the β -transus temperature to avoid significant microstructural coarsening (in the titanium industry, β -annealing has been largely replaced by recrystallisation annealing for the same reason). Such heat treatments do not change the prior- β grain structures (which has been well documented). Therefore, the ultrasound-induced microstructural features, including the fine, equiaxed prior- β grains and substantially weakened texture, will persist after normal post-AM heat treatments. In other words, the benefits of ultrasound-assisted AM observed in this study will be retained after such post-AM heat treatments.

As regards stress relief annealing, we made good effort previously to quantify the difference in residual stress in simple tensile sample of Ti-6Al-4V built by both SLM and directed energy deposition under different conditions using a dedicated X-ray diffractometer at the University of Melbourne (both martensitic and fully α - β lamellar Ti-6Al-4V). The scatter in the residual stress data obtained excluded us from making any firm conclusion.

From a practical application perspective, for as-built fully lamellar α - β Ti-6Al-4V (which is the case of this study), in fact, the effect of stress relief annealing on tensile properties is often small to negligible (although it can be important to avoid pronounced distortion of large parts before removal from the substrate). This is consistent with the literature data. **For instance, in two independent studies [9-10], the application of stress relief annealing (~600 °C for 2-4 hr) to laser directed energy deposited Ti-6Al-4V changed their tensile properties by just ~1-2%. This percentage change is within the scatter of the tensile data of this work (~1-2% for strengths and ~10-15% for ductility).** Thank you for this suggestion. We will investigate the effect of ultrasound on the residual stress in intricate samples in the future. It would be another great benefit if we could quantify that ultrasound can noticeably reduce the residual stress in the as-built Ti-6Al-4V samples through a systematic study.

Throughout the revised manuscript, we have indicated that the tensile properties of the additively manufactured Ti-6Al-4V studied in the present work were for the as-built conditions with machined surfaces. Based on the current literature information [8, 12, 17], we have made the following addition to the revised manuscript:

“We note that post-AM heat treatments below the β -transus temperature including hot isostatic pressing are often applied to Ti-6Al-4V for improved strength-ductility combinations and property consistency^{8,9}. Such heat treatments do not change the prior- β grain structures^{7,10,17}. Hence, the effect of the ultrasound-induced microstructural changes, i.e., the equiaxed prior- β grains, reduced prior- β grain size and substantially weakened texture, on mechanical properties shown in Fig. 5, is expected to survive after common post-AM heat treatments.”

Comment #3: Next is fatigue. You mention off-hand that the equiaxed Ti parts will have better fatigue, but you don't show this. It would be notable if the fatigue of Ti parts using the sonotrode method is superior to Ti parts made other ways when no heat treatments are applied. Since the parts you show are made through DED, they will have to be machined anyways, due to the low resolution of the DED printer. Will your method produce better properties once the parts are machined?

Response: Thank you and we agree that we have made an offhand claim without necessary experimental backup. We have now removed all the conjectures about the fatigue issue from the revised manuscript. This comment deals with the influence of the much-weakened texture and refined prior- β grains on the fatigue performance of the samples with and without post-AM surface finish by machining. It is a practically important question. We will need to investigate this in detail in the future with a large batch of as-built fatigue samples, with and without machining, and probably, with and without hot isostatic pressing as well. For the time being, this manuscript will be restricted to the solidification microstructure and the resulting tensile properties in the as-built state for non-fatigue demanding applications.

Comment #4: Next is alloys: You mention that the process can be used for other alloy systems to achieve refined grains. Why not show this in some other alloy to generalize the process? Print some steel and show that the grains are different. Does this process work across alloy systems?

Response: We really appreciate this comment. We have designed a custom-made new sonotrode for AM of Inconel 625, another important commercial alloy that shows strong columnar grain structures by AM. Fig. 6a-d below summarizes the exciting results obtained from AM of Inconel 625 with and without ultrasound. As expected, ultrasound-assisted AM

consistently converted the strong columnar grains of Inconel 625 into fine, equiaxed grains (with a well-weakened texture).

Fig. 6a-d Additive manufacturing of Inconel 625 with and without high-intensity ultrasound. **a, b** Inverse pole figure maps along the build direction (z) for the γ phase in samples **a** without and **b** with ultrasound. **c, d** $\{001\}$ contoured pole figures of the γ phase in samples **c** without and **d** with ultrasound. The contoured pole figure in **c** was obtained using the larger area EBSD data provided in Supplementary Fig. 5. Scale bars, 250 μm .

In the revision, we have added an extra Main Figure (Fig. 6a-d, new Inconel 625 results) and the following discussion:

“To test the generality of our approach, we have similarly applied high-intensity ultrasound to AM of Inconel 625 using a custom-made stainless steel 4140 sonotrode (see details in the ‘Methods’ section). The sample fabricated without ultrasound exhibits columnar primary γ grains of 500 μm in length and 150 μm in width with a strong $\langle 001 \rangle$ texture (Fig. 6a, c and Supplementary Fig. 5). In contrast, the application of ultrasound produces predominately equiaxed primary γ grains of only a few microns in size (much finer than Ti-6Al-4V) with a near random crystallographic texture (Fig. 6b, d). This confirms the generality of the ultrasonic approach for AM of different metallic materials.”

Comment #5: Lastly, the paper seems to be lacking discussion about using this technique to produce parts with microstructural gradings. If the sonotrode can be turned on and off during a print, couldn't you produce a part with both equiaxed and columnar grains? Why didn't you do that? That would have been an amazing demonstration of this technique. Perhaps you should make a few more Ti samples and add this to the paper.

Response: We really appreciate this exciting suggestion! We have designed specific experiments to answer this interesting question. Fig. 6e below shows the exciting grain structures obtained from switching on and off the ultrasound during AM of Inconel 625. The correspondence is straightforward, i.e., well grain refined when the ultrasound is switched on while strong columnar grains return when the ultrasound is switched off. *We would like to thank you again for this fantastic suggestion!*

We have added the following to the revised manuscript:

“To further showcase the capability of our approach for solidification control during AM, we fabricated a microstructurally graded Inconel 625 sample that exhibits an alternating columnar/equiaxed/columnar grain structure along its build height, as shown in Fig. 6e. This was achieved by simply turning on and off the high-intensity ultrasound during AM. The approach thus also offers an alternative means of fabricating graded grain structures during AM.”

Fig. 6e The ultrasound on-and-off effect on grain refinement of Inconel 625 during AM. Inverse pole figure map along the build direction (z) of an Inconel 625 sample fabricated by turning the ultrasound on and off during AM. Scale bar, 250 μm .

Comment #6: Overall, I thought the paper was very interesting in terms of what was achieved in Ti. However, the process was demonstrated for DED Ti and the discussion claims that it could be used in multiple alloys across multiple printers. I do not think that is supported. The authors need to either add additional details and experiments to justify their claims or they need to clearly point out the limitations of the technique to their experimental setup.

Response: Thank you for this important comment. This relates to two of your previous comments. We have discussed the limitations of the technique (please refer to our response to your comment #1). We have also demonstrated this technique for additive manufacturing of Inconel 625 (please see our response to your comment #4), as well as the on-and-off effect of ultrasound (please see our response to your comment #5). Thank you again for this excellent review!

Reviewer #3:

Comment #1: A direct proof of the hypothesis would be necessary. The paper is still at the stage of hypothesis formulation. It is likely that ultrasound could generate cavitations that enhance nucleation. There is no conclusive and direct evidence in the manuscript proving this hypothesis. The calculation of ultrasound intensity helps, but not sufficient to support the conclusion.

Response:

Thank you for this comment. We would like to take this opportunity to emphasize that the novelty or focus of this work is the method described to control solidification microstructure during additive manufacturing (AM), namely (i) to enable columnar-to-equiaxed transition without changing alloy composition, (ii) to identify the resulting changes in texture, and (iii) to determine the influence of both the much weakened texture and the refined prior- β grains on tensile properties. The approach demonstrated on both Ti-6Al-4V and Inconel 625, including the *on-and-off* effect, “*will likely change the way that future metal 3D printers are built*” (Reviewer #2). It addresses a timely and important issue in the field and offers new opportunities for microstructure manipulation (e.g., microstructural gradings, see the on-and-off effect demonstrated earlier in Fig. 6e). Our response to the next question provides more details and discussion. Please see below.

Comment #2: The authors have previously published works on ultrasound induced grain refinement. What is the new physics when applying this technique in laser fusion AM process? Does the high G in AM process change any of the characteristics comparing with previous studies? It is not clear how this paper is different from previous published works in terms of grain refinement mechanism.

Response: We appreciate these questions and first respond to the question about the high G .

The effect of the high G in AM processes:

Previous studies by the authors dealt only with conventional solidification conditions with a low temperature gradient (G). Under such conventional casting conditions, Ti-6Al-4V, Inconel 718 and other commercial alloys typically solidify as an equiaxed grain structure [11-12]. In contrast, the high G in the AM processes stimulates strong columnar grains in these alloys, which makes it considerably challenging to disrupt the columnar growth without

changing alloy composition. The influence of the high G on the columnar development in additively manufactured alloys has been well identified and described through the G - V (V : growth velocity) diagram [13-14]. The high G renders a high cooling rate ($G \times V$). Recent research has shown that the efficiency of ultrasonic grain refinement increases with increasing cooling rate up to $10^4 \text{ }^\circ\text{C s}^{-1}$ investigated [15]. In that regard, the high G during AM may have assisted in the grain refining efficiency.

New physics and grain refinement mechanism compared with previous work:

Thank you for this insightful question. There is at least one distinct difference between ultrasonic grain refinement during AM and conventional solidification. This is related to the very small melt pool during AM. In this study, the laser beam size used was 0.61 mm. The melt pool size is typically ~30% larger than the beam size, which gives ~0.8 mm. For conventional ultrasonic grain refinement in a large volume of melt, the basic principles are generally clear, i.e., cavitation is essential for producing a large number of nuclei or crystallites (up to four different mechanisms may be operative [16-17]) [18-22], while acoustic streaming is important for the distribution of these crystallites from the sonotrode region to the rest of the melt [23-24]. Due to the very small melt pool (~0.8 mm), the distribution effect of the acoustic streaming on grain refinement during AM can be assumed to be minimal.

The cavitation threshold is determined by the surface tension of the melt. For molten Ti-6Al-4V, its surface tension (σ) above the liquidus temperature (1655 $^\circ\text{C}$) has been systematically measured and obeys the following relationship with melt temperature (T) [25]:

$$\sigma = 1.52 - (T - 1655) \times (5.52 \times 10^{-4}) \text{ (in N m}^{-1}\text{)}$$

The melt pool temperature of Ti-6Al-4V during directed energy deposition has been measured and the temperature at the centre can reach 2500 $^\circ\text{C}$ [26]. Accordingly, the surface tension of the Ti-6Al-4V melt during AM is about 1.05 N m⁻¹.

Measuring cavitation in molten metals is challenging and we are not aware of any experimental measurement in molten Ti and Ti alloys. However, Eskin has systematically measured cavitation in commercially pure molten Al [17]. The surface tension of molten Al at 710 $^\circ\text{C}$ was measured to be ~0.9 N m⁻¹ [27], which is similar to the surface tension of molten

Ti-6Al-4V (1.05 N m^{-1}) in the melt pool during directed energy deposition. The ultrasonic conditions used in this study (ultrasonic amplitude: $30 \mu\text{m}$, frequency: 20 kHz) are included in those used by Eskin [17]. Eskin [17] investigated the effect of ultrasonic amplitude (2, 5, 10, 15, 20, 30 and $40 \mu\text{m}$) on cavitation in molten Al and identified that cavitation was incipient when the amplitude was increased from $2 \mu\text{m}$ to $5 \mu\text{m}$ and became fully developed when the amplitude reached $10 \mu\text{m}$ and beyond. As shown by Eq. 1 in the manuscript, the ultrasonic intensity is proportional to the square of amplitude (A). The $30\text{-}\mu\text{m}$ amplitude applied in this study is 9 times the ultrasonic intensity required for the generation of fully developed cavitation in molten Al. As pointed out earlier, the surface tension of molten Ti-6Al-4V in this study is similar to that of the molten Al investigated by Eskin [17]. In other words, the ultrasonic conditions employed in this study are well above the threshold required for cavitation in molten Ti-6Al-4V during directed energy deposition.

Kinetically, the directed energy deposition process used in this study offers far more than sufficient time for cavitation to occur in the melt pool. As specified earlier, the melt pool size in this study is about 0.8 mm . The laser scanning speed used was 600 mm min^{-1} . The melt pool survival time is equivalent to the time for the laser to travel across the melt pool, which is 0.08 s . On the other hand, the time required from bubble formation to bubble implosion in liquid metals is $\sim 30 \mu\text{s}$ (0.00003 s), measured by *in situ ultrafast high-resolution synchrotron X-ray imaging* [22] (please refer to our response to the next comment, comment #3, on the kinetics of cavitation in molten metals). Hence, the melt pool survival time is far more than sufficient for cavitation to occur in the melt pool.

The above response with detailed references has been included in the Supplementary Materials (Supplementary Note 1).

In summary, thanks to your comment #1 and comment #2, we have removed the entire paragraph on cavitation in the original discussion and replaced it with the following:

“For conventional ultrasonic grain refinement in a large volume of melt, the basic grain refinement mechanisms are generally clear, i.e., cavitation is essential for the production of a large number of nuclei or crystallites (up to four different mechanisms may be operative^{37,43})^{34,60-62}, while acoustic streaming is important for their distribution from the sonotrode region to the rest of the melt^{63,64}. Due to the very small melt pool ($\sim 0.8 \text{ mm}$) during AM, the distribution effect of acoustic streaming on grain refinement can be assumed to be

minimal, because the entire small melt pool is effectively ultrasonicated. As detailed in Supplementary Note 1, the DED process employed in this study offers far more than sufficient time for cavitation to occur (~ 0.00003 s) in the melt pool. On the other hand, the ultrasonic intensity employed in this study far exceeds the threshold for cavitation in molten Ti-6Al-4V, estimated according to the surface tension of molten Ti-6Al-4V (~ 1.05 N m⁻¹) during AM, compared with that of molten Al (~ 0.9 N m⁻¹) and the detailed measurements of cavitation in molten Al by Eskin³⁷. In that regard, cavitation can be assumed to be the predominant reason for the grain refinement observed.”

Comment #3: Any effects of the ultrasound on the deposited powders? Does it induce movement of powders? Any effects on the final porosity comparing with that without ultrasound?

Response: Thank you for these important comments.

Effect on deposited powders and movement of powders

During AM by laser-based directed energy deposition, the Ti-6Al-4V powder (45-90 μ m in size) is sprayed from a nozzle directly into the small melt pool (~ 0.8 mm wide) and is melted immediately. The nozzle is well above the melt pool and has no contact with the sonotrode. Therefore, the ultrasound does not affect the solid powder particles. Please also refer to our response to Reviewer #2's comment #1 about the potential limitation of this technique.

Effect on the final porosity

Thank you again for this inspiring comment. It is a logical concern when cavitation has been assumed to be the predominant mechanism for grain refinement. A recent detailed study of ultrasound-induced cavitation in a molten Bi-Zn alloy using in situ ultrafast high-resolution synchrotron X-ray imaging has clearly shown that it takes only ~ 30 μ s (0.00003 s) from the formation of a small bubble to its collapse (cavitation) [22]. The imploded bubble produced many tiny bubbles or bubble fragments (smaller than 1 μ m) until they completely disappeared or were no longer detectable. No increase in porosity was observed. In fact, Eskin has already indicated that, in addition to grain refinement, ultrasonication also leads to reduced porosity at the same time [17]. We have examined and compared several polished cross-sections of the Ti-6Al-4V samples with and without ultrasound. Supplementary Fig. 6 below shows representative observations of each sample. As expected, the results revealed no obvious

changes to the porosity by ultrasound (the total porosity on the polished cross-sections of both samples is ~0.7-0.9 area%) and are generally consistent with Eskin's observations.

We have added a Supplementary Fig. 6 and also the following text to our revised manuscript: "Detailed optical microscopy analysis has revealed that the porosity on the polished cross-sections of the Ti-6Al-4V samples with and without ultrasound was similar, in the range of 0.7-0.9 area% (see Supplementary Fig. 6)."

Supplementary Fig. 6 The effect of ultrasound on the porosity in the AM-fabricated Ti-6Al-4V samples. **a, b** Optical microscopy images of the polished cross-sections of the samples **a** without and **b** with ultrasound. The porosity in both samples is ~0.7-0.9 area%. Scale bars, 250 μm .

References:

1. C. Cayron: *J. Appl. Crystallogr.*, 2007, vol. 40, pp. 1183-88.
2. B. J. Hayes, B. W. Martin, B. Welk, S. J. Kuhr, T. K. Ales, D. A. Brice, I. Ghamarian, A. H. Baker, C. V. Haden, D. G. Harlow, H. L. Fraser and P. C. Collins: *Acta Mater.*, 2017, vol. 133, pp. 120-33.
3. D. G. Lee, S. Lee, C. S. Lee and S. Hur: *Metall. Mater. Trans. A*, 2003, vol. 34, pp. 2541-48.
4. I. Sen, S. Tamirisakandala, D. Miracle and U. Ramamurty: *Acta Mater.*, 2007, vol. 55, pp. 4983-93.

5. X. Tan, Y. Kok, Y. J. Tan, M. Descoins, D. Mangelinck, S. B. Tor, K. F. Leong and C. K. Chua: *Acta Mater.*, 2015, vol. 97, pp. 1-16.
6. W. Xu, M. Brandt, S. Sun, J. Elambasseril, Q. Liu, K. Latham, K. Xia and M. Qian: *Acta Mater.*, 2015, vol. 85, pp. 74-84.
7. Y. M. Ren, X. Lin, X. Fu, H. Tan, J. Chen and W. D. Huang: *Acta Mater.*, 2017, vol. 132, pp. 82-95.
8. M. J. Bermingham, L. Nicastro, D. Kent, Y. Chen and M. S. Dargusch: *J. Alloy. Comp.*, 2018, vol. 753, pp. 247-55.
9. B. Baufeld, E. Brandl and O. van der Biest: *J. Mater. Process. Tech.*, 2011, vol. 211, pp. 1146-58.
10. N. Chekir, Y. Tian, R. Gauvin, N. Brodusch, J. J. Sixsmith and M. Brochu: *Mater. Sci. Eng. A*, 2018, vol. 724, pp. 335-47.
11. M. J. Bermingham, S. D. McDonald, M. S. Dargusch and D. H. StJohn: *J. Mater. Res.*, 2008, vol. 23, pp. 97-104.
12. L. Nastac and D. M. Stefanescu: *Metall. Mater. Trans. A*, 1996, vol. 27, pp. 4075-83.
13. P. A. Kobryn and S. L. Semiatin: *J. Mater. Process. Tech.*, 2003, vol. 135, pp. 330-39.
14. N. Raghavan, R. Dehoff, S. Pannala, S. Simunovic, M. Kirka, J. Turner, N. Carlson and S. S. Babu: *Acta Mater.*, 2016, vol. 112, pp. 303-14.
15. J. G. Jung, T. Y. Ahn, Y. H. Cho, S. H. Kim and J. M. Lee: *Acta Mater.*, 2018, vol. 144, pp. 31-40.
16. M. Qian, A. Ramirez and A. Das: *J. Cryst. Growth*, 2009, vol. 311, pp. 3708-15.
17. G. I. Eskin and D. G. Eskin: *Ultrasonic Treatment of Light Alloy Melts*, 2nd ed., CRC Press, Boca Raton, 2014.
18. J. W. Mi, D. Y. Tan and T. L. Lee: *Metall. Mater. Trans. B*, 2015, vol. 46, pp. 1615-19.
19. D. Y. Tan, T. L. Lee, J. C. Khong, T. Connolley, K. Fezzaa and J. W. Mi: *Metall. Mater. Trans. A*, 2015, vol. 46a, pp. 2851-61.
20. F. Wang, I. Tzanakis, D. Eskin, J. Mi and T. Connolley: *Ultrason. Sonochem.*, 2017, vol. 39, pp. 66-76.
21. F. Wang, D. Eskin, J. W. Mi, C. N. Wang, B. Koe, A. King, C. Reinhard and T. Connolley: *Acta Mater.*, 2017, vol. 141, pp. 142-53.
22. B. Wang, D. Y. Tan, T. L. Lee, J. C. Khong, F. Wang, D. Eskin, T. Connolley, K. Fezzaa and J. W. Mi: *Acta Mater.*, 2018, vol. 144, pp. 505-15.

23. G. Wang, P. Croaker, M. Dargusch, D. McGuckin and D. StJohn: *Comp. Mater. Sci.*, 2017, vol. 134, pp. 116-25.
24. G. Wang, Q. Wang, N. Balasubramani, M. Qian, D. G. Eskin, M. S. Dargusch and D. H. StJohn: *Metall. Mater. Trans. A*, 2019, vol. 50, pp. 5253-63.
25. R. Aune, L. Battezzati, R. Brooks, I. Egry, H. J. Fecht, J. P. Garandet, K. C. Mills, A. Passerone, P. N. Quested, E. Ricci, S. Schneider, S. Seetharaman, R. K. Wunderlich and B. Vinet: *Microgravity Sci. Technol.*, 2005, vol. 16, pp. 11-14.
26. G. J. Marshall, W. J. Young, S. M. Thompson, N. Shamsaei, S. R. Daniewicz and S. Shao: *JOM*, 2016, vol. 68, pp. 778-90.
27. I. F. Bainbridge and J. A. Taylor: *Metall. Mater. Trans. A*, 2013, vol. 44a, pp. 3901-09.

REVIEWERS' COMMENTS:

Reviewer #1 (Remarks to the Author):

The authors present a novel way of controlling the microstructure (grain/texture) of DED additively manufactured materials by means of applying ultrasonic waves. They demonstrate the effect and explain the theory, and validate the effect on a second material.

This is a response to previous comments by reviewers. The authors have appropriately and completely responded to the reviewers comments.

I think that the paper stands on its own merit, and will represent a strong contribution to the materials science community in general and the additive manufacturing community specifically.

-Pete Collins

Reviewer #2 (Remarks to the Author):

The authors have done a wonderful job with adding additional experiments to the paper to make it stronger. If we look back at the original paper, there was one demonstration of grain refinement using the ultrasound and a bunch of unsupported claims about the applicability of the process. Now, the revised paper has a demonstration that the process works also in Inconel (which shows broad applicability) and there is a demo of how the microstructure can be functionally graded by turning on and off the ultrasound. Even though the technique appears to have the best suitability for DED printing, the authors have shown that microstructures can be obtained that can't be made any other way. I think they have made the paper far better by taking the time to do additional experiments and I think they will be rewarded by having a paper that will be widely cited and possibly used in many future applications. I commend the authors for making such an effort to improve the paper.

Regards,
Douglas Hofmann

Reviewer #3 (Remarks to the Author):

The authors have addressed the issues previously raised. It is suggested to accept as it is.